# Signature Activation: A Sparse Signal View for Holistic Saliency

**Jose Roberto Tello Ayala** [1]   **Akl C. Fahed** [2][3]   **Weiwei Pan** [1]   **Eugene V. Pomerantsev** [3]   **Patrick T. Ellinor** [2][3]
**Anthony Philippakis** [2]   **Finale Doshi-Velez** [1]

## Abstract

The adoption of machine learning in healthcare calls for model transparency and explainability. In this work, we introduce Signature Activation, a saliency method that generates holistic and class-agnostic explanations for Convolutional Neural Network (CNN) outputs. Our method exploits the fact that certain kinds of medical images, such as angiograms, have clear foreground and background objects. We give theoretical explanation to justify our methods. We show the potential use of our method in clinical settings through evaluating its efficacy for aiding the detection of lesions in coronary angiograms.

## 1. Introduction

Machine learning models are increasingly deployed in high stakes applications like healthcare (Gulshan et al., 2016). For these systems to be used in clinical settings, it is imperative that human experts understand and agree with the models' decision process. Though powerful, the black-box nature of most complex machine learning models (e.g. neural network models) hampers their widespread adoption as they remain difficult to interpret.

In the context of images, saliency and class activation maps are explanation methods that interpret the classifications of CNNs. These methods generate pixel maps that highlight, within the original input image, regions that contribute significantly to the classification of a specific class. Current saliency and class activation maps underscore the regions that maximize the probability output associated with a particular class. While sometimes valuable, these types of

approach to identifying the most relevant parts of an image have two major issues: First, if there are multiple probable classes, only the most probable is explained. Second, by focusing on the most relevant pixels, these approaches are incomplete: aspects of the image that may assist in interpreting the classification but are not ranked as highly relevant may get pruned.

In medical imaging settings it is imperative to have the most accurate and faithful explanations to evaluate a machine learning model. A biased explanation can be particularly problematic as we not only want to verify if the model is looking at the most relevant features that contribute towards the decision, but we also need to evaluate if the method is aligned with how clinicians perform the evaluation.

In this work, we propose a novel method, Signature Activation, that more faithfully captures CNN model decision making for image classification tasks. In particular, our method produces class-agnostic explanations – highlighting regions in the input image that informs the model's entire prediction (i.e. classification probabilities for all classes). Inspired by notions of saliency in classical computer vision, object detection, and utilizing the theory of sparse signal mixing, our method approximates the way models extract of relevant information in the frequency domain of learned representations. Moreover, our approach only requires access to the model and not its gradients.

Our Signatature Activation approach localizes the relevant foreground information through object detection in the context of both natural images and the task of detecting lesions in coronary angiography (Figure 1). Our method produces explanations that are both more faithful to the model as well better aligned with clinical insights. We also characterize the conditions under which our (sparse) approach is desirable.

## 2. Related Works

Saliency type explanations have been previously used to verify the output classification of models to detect diseases or lesions in medical computer vision such as diabetic retinopathy (Sayres et al., 2019), tumor detection in the brain through MRIs (Windisch et al., 2020), and detection

---

[1] John A. Paulson School of Engineering and Applied Sciences, Harvard University, Cambridge, Massachusetts [2] Broad Institute of Massachusetts Institute of Technology and Harvard, Cambridge, Massachusetts [3] Division of Cardiology, Massachusetts General Hospital, Harvard Medical School, Boston, Massachusetts. Correspondence to: Jose Roberto Tello Ayala <jtelloayala@g.harvard.edu>.

*Workshop on Interpretable ML in Healthcare at International Conference on Machine Learning (ICML)*, Honolulu, Hawaii, USA. 2023. Copyright 2023 by the author(s).

of pneumonia on COVID-19 patients with chest CT scans (Harmon et al., 2020). These types of explantion methods can broadly be described as Class-Discriminative and Class Agnostic.

**Class-Discriminative Saliency Methods** There are a large number of approaches for generating class saliency activation maps. Methods such as backpropagation (Simonyan et al., 2014), Integrated Gradients (Sundararajan et al., 2017), CAM (Zhou et al., 2015), Grad-CAM (Selvaraju et al., 2017), and Grad-CAM++ (Chattopadhay et al., 2018) operate by taking gradients with respect to a particular class. In addition to issues with using gradients (such as noise and gradient saturation), these offer only partial explanations related to a particular class. Gradient-free algorithms such as Ablation-CAM (Desai & Ramaswamy, 2020) and Score-CAM (Wang et al., 2020) have better stability because they do not use gradients. However, they too are designed to focus information related to a specific classification. They are also more computationally expensive than the approaches that use gradients. Our approach, Signature Activation is computationally efficient whilst being gradient free; unlike these class-discriminative approaches, it also includes a more complete array of pixels that the model use for classifying.

**Class Agnostic Saliency Methods** Non-class discriminative methods such as CNN-Fixations (Mopuri et al., 2019) and Eigen-CAM (Muhammad & Yeasin, 2020) look more holistically at what the model fixates on for producing the final decision, rather than focusing on pixels relevant only to a particular class. Nonetheless, CNN-Fixations faces significant challenges, not only in terms of substantial computational costs, but also due to the complex and potentially prohibitive manipulation required when working with pre-trained models. On the other hand, Eigen-CAM is constrained by its ability to abstract only linear representations of the data, thereby limiting its capability to capture more complex relationships. As previously mentioned, Signature Activation is computationally efficient, requires only access to the layers of the model, and is able to capture non-linear relationships of the data.

## 3. Background

It is well known that CNNs learn hierarchical representations of the training data (LeCun et al., 2015), with deeper layers abstracting higher semantical features of the data. Previous saliency methods such as Grad-CAM, Grad-CAM++, Score-CAM, and Eigen-CAM have exploited the semantic information captured in the last convolutional layers to generate attribution heatmaps, either through backpropagation, maximizing the score of the highest predicted class, or through Principal Component Analysis. These methods

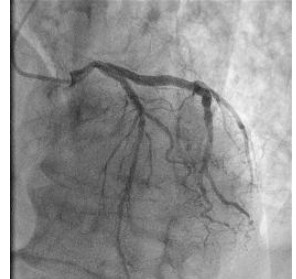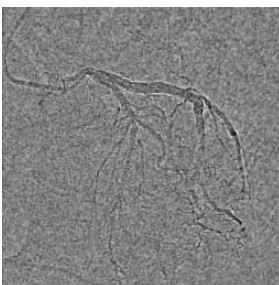

*Figure 1.* One frame taken from a Left Coronary Angiogram. On the left, the frame is presented without modifications. On the right, the background is suppressed by computing the image signature of the original image and then recovered through the Inverse Discrete Cosine Transform. The background is effectively suppresed.

utilize the learned filters of the last convolutional layer to generate class activation maps. These maps are later superimposed over the original image to highlight the regions of interest for the classification. For developing Signature Activation, we also seek to utilize the last convolutional layer to generate heatmaps, while employing the properties of the forward pass.

## 4. Problem Setting and Classifier: Invasive Coronary Angiograms

Invasive coronary angiograms, or simply coronary angiograms, are a diagnostic procedure that involves the insertion of a catheter into the body often through the wrist or groin, threading it up to the coronary artery where a contrast dye is injected into the patient's bloodstream. As the dye travels through the heart's arteries it makes them visible under X-ray imaging (Kern et al., 2019). This allows physicians to identify any blockages and lesions that may be impairing the heart's performance.

When AI tools are used to assist with lesion detection, explanations can help calibrate trust in the classification. An effective explanation for a model trained to identify coronary lesions should mirror the diagnostic process undertaken by a clinician. It must convincingly demonstrate the model's capacity to meticulously examine the entirety of the arterial structure, accurately distinguishing it from surrounding anatomy. Furthermore, it should substantiate the model's capability to pinpoint the lesion's location.

For example, in a lesion detector classifier, knowing if the model is able to detect the lesions is critical. It is also required to understand if the model is overall evaluating only the regions of interest, without fixating on the background or learning human annotations in the image (Kaufman et al., 2011; Yagis et al., 2021).] In our work, we trained two binary lesions classifiers, for the right and the left coro-

nary artery accordingly, through transfer learning with the MoviNets architecture for video classification (Kondratyuk et al., 2021).

# 5. Failures of Traditional Explanation Methods

A common approach to problems like explaining lesion detection above involve creating class-discriminative saliency maps. These explanations highlight the areas of an image that contributes the most towards the output probability for the given category of interest. They are usually computed with respect to the label that outputs the highest probability. The generated explanation can allow the end user to understand what features of a particular class are the most relevant in the decision making for the model. Saliency maps can also aid in the debugging and validation of models (Adebayo, 2022; Montavon et al., 2018). However, class-based discrimination methods do not provide an explanation that considers the whole context of what the model detects during the forward pass, not being faithful to how the model operates for taking the decision. For example, a neural network classifier $f$ outputs the probabilities for every class at evaluating the image during the forward pass computation $f(I)$. While class-discriminative methods only focus on the direction of the steepest change with respect to the loss a particular at a time. With gradients methods this entails computing $\frac{\partial L_c(f)}{\partial I}$, involving the backpropagation operation which is not used to generate the output probabilities.

One set of issues arise because, by design, these explanations are developed to highlight regions of importance for just a particular label. By only focusing on one class at a time, the explanation does not capture the full spectrum of all the possible regions of relevance, hence limiting its scope of interpretation. This leads to a potential lack of comprehensive understanding of other classes and the relationships between them which could impact overall model robustness and performance.

Let us consider a specific case: suppose we have a classifier that only learns to detect moderate to severe lesions. Suppose that we input an angiogram of a patient with both a mild and moderate lesion in different locations. A class-discriminative explanation that highlights the most relevant features for this patient will highlight the location of the moderate lesion only, even though there is also a mild lesion also present. The healthcare provider may then dismiss the mild lesion as a potential object of concern, or miss it entirely. Even if the classifier had multiple outputs for mild, moderate, and severe lesions, the above problem could still happen. For example, if the moderate lesion had the highest class probability, then an interpretation based on that class would emphasize only the moderate lesion in the image, ignoring other regions that might also be impacting

the decision process (probabilities associated with the mild lesion). In instances where an image display occurrences of multiple classes, it is important to know if the model also detects the other objects or regions of interests, as most classifiers output all the class probabilities instead of just the highest one. As models can also make mistakes, an explanation that only provides evidence for a falsely positive classification can also lead to unneeded interventions for the patient such as a coronary stent. Such confirmation biases are well-documented in the literature (Adebayo et al., 2018).

Besides leaving out information that might support other classes, the sparsity induced by class-discriminative saliency maps can hide information about what the model is using for even the highest probability class. Specifically, cardiologists performing coronary angiograms often look at the whole vessel to determine the presence and location of the lesion when the dye fills up the artery. Our trained models do the same. However, in Figure 3, we can appreciate how Grad-CAM only highlights patches where the lesions are located—even though other parts are being used. Indeed, our proposed method faithfully suggests that the model fixates on the entire vessel to output its decision.

In Figure 2, we show another example of this effect based on an example from a ConvNeXtLarge architecture (Liu et al., 2022) pre-trained on Imagnet (Russakovsky et al., 2015) on explaining an image with both "tiger cat" and "remotes". Grad-CAM only highlights patches of the image where the tiger cats are located, while Imagenet contains labels for both "tiger cat" and "remote" and the classifier outputs probabilities for both. In contrast, our proposed method detects not only the cats but it also highlights the remotes, displaying features relevant for both. It also shows that the model detects the cats' paws which are not considered as regions of importance by Grad-CAM, although a human could consider them as relevant features of a cat. It differentiates the points of attention on what the model focuses on while not underscoring the background.

# 6. A Novel Explanation Method: Signature Activation

As noted in Section 3, none of the former algorithms exploit the properties of the forward pass to generate their attribution mask. As these methods do not make use of the properties of the forward pass, their explanations are unfaithful to the decision making process that the model takes. We develop a novel saliency method that takes advantage of the foreground-background sparsity of images and how that sparsity is propagated through the forward pass in the CNN. Our intuition draws from the fact that Papyan et al. (2017) prove that "the forward pass is guaranteed to recover an estimate of the underlying representations of an input signal, assuming these are sparse in a local sense" (p. 3).

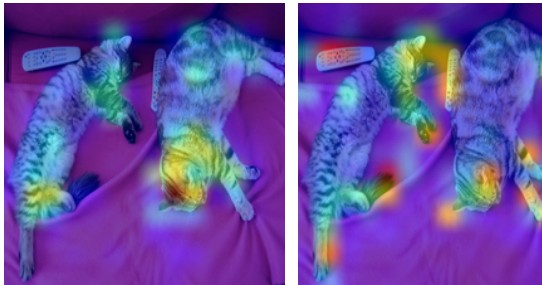

*Figure 2.* Image that contains instances from two classes from Imagenet, Tiger Cat and Remote, with two different heatmaps. On the left, the heatmap overlay corresponds to the features that are highlighted by Grad-CAM with the highest probability corresponding to *tiger cat*. On the right, the heatmap overlay corresponds to the features that are highlighted by Signature Activation. Both produced with the second to last layer convolutional layer of ConvNeXtLarge. Signature Activation detects both of the remotes as well as the cats.

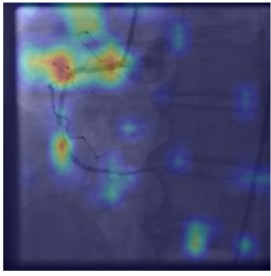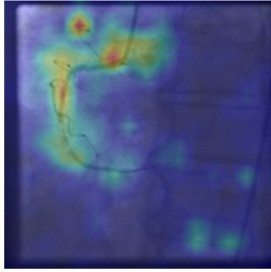

*Figure 3.* One frame taken from a Right Coronary Angiogram with a lesion, with two different heatmaps. One the left, the heatmap overlay corresponds to the features that are highlighted by Grad-CAM. On the right, the heatmap overlay corresponds to the features that are highlighted by Signature Activation. Both produced with MoviNets.

Natural images tend to have a clear foreground-background separation. The subjects or objects remain the primary area of in interest in most computer vision tasks with the scene usually being relegated (Liu et al., 2018). Let $I \in \mathbb{R}^{n \times n}$ be a black and white image. Assuming the signal can be decomposed as foreground $f \in \mathbb{R}^{n \times n}$ and background $b \in \mathbb{R}^{n \times n}$ we can write $I$ as

$$I = f + b. \tag{1}$$

The problem of separating the foreground and background known as *figure-ground segmentation* is a well studied concept both from the cognitive science and computer vision communities (Frintrop et al., 2010). The above decomposition can be illustrated in the context of Coronary Angiograms. In an angiogram there is a clear background and foreground where, the foreground $f$ represents the highlighted artery by the dye and the background $b$ corresponds

to the *gray* area were there are no highlighted vessels. In Figure 1 we can appreciate an instance where the background in a Left Coronary Angiogram is suppressed.

Classical computer vision distinguished objects from images via digital image processing techniques such as contour detection, feature extraction, and spectral analysis (Gonzalez & Woods, 2018). The success of Neural Networks for object detection and classification tasks has shifted most of the efforts of the computer vision community towards CNNs and Transformer based architectures (Girshick et al., 2013; Redmon et al., 2016; Dosovitskiy et al., 2021). However the ground work and theory of image processing can still be applied to better understand and generate faithful explanations for CNNs.

Following Hou et al. (2012) we assume that an image $I \in \mathbb{R}^{n \times n}$ can be decomposed as Equation 1 where $f$ is sparse in the standard spatial basis and $b$ is sparsely supported in the basis of the Discrete Cosine Transform (DCT) (Ahmed et al., 1974), i.e. $b = Cx$ where $C$ is an orthogonal matrix where each column corresponds to a DCT basis vector and $x$ is sparse. We seek to isolate the foreground from the background. In order to do so we introduce the following tool:

**Definition 6.1.** Let $I \in \mathbb{R}^{n \times n}$ be a black an white image, the image Signature of $I$, denoted as $\mathrm{Sig}(I)$, is defined as the sign of the coefficients of the Discrete Cosine Transform:

$$\mathrm{Sig}(I) = \mathrm{sign}(\mathrm{DCT}(I)).$$

When looking at only the signs of the DCT coefficients, we essentially extract a form of high-level, structural information about the image. By computing the Inverse Discrete Cosine Transform (IDCT) of the image signature, we obtain a reconstructed image that approximately isolates the support of $f$. Based on the Uniform Uncertainty Principle (Candes & Tao, 2006), Hou et al. (2012) proved in Proposition 1, please reference to Appendix A for the precise statement of the theorem, that in expectation the cosine similarity between $\mathrm{IDCT}(\mathrm{Sig}(f))$ and $\mathrm{IDCT}(\mathrm{Sig}(I))$ is greater than $0.5$. Cosine similarity of -1 indicates no resemblance between the inputs, 0 means decorrelation between the inputs, and 1 signals total agreement between the inputs. This translate to having in expectation a fair extraction of the foreground by computing $\mathrm{IDCT}(\mathrm{Sig}(f))$. Notice that the above might not occur for images that do not have an object of interest or a clear foreground-background separation.

Given that angiograms have a direct focal object of attention and backdrop, the suppression of the background through the Image Signature is effective as can be seen in Figure 1. The main driving question for the presented method was: do CNNs only look at objects of interest during the forward pass? Papyan et al. (2017) showed that the forward pass approximately solves a multi-layer convolutional sparse

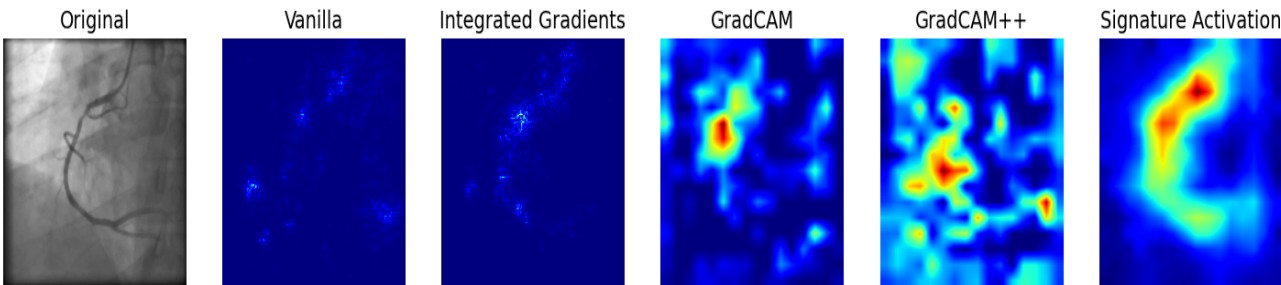

*Figure 4.* One frame from a Right Coronary Angiogram and different masks generated by different methods. Signature Activation detects the whole vessel and provides an approximate area for where the lesion is located.

coding model (ML-CSC). As the forward pass computation approximately estimate the sparse local patches that make up the image signal, we hypothesize that both, the spatial sparsity of the foreground as well as the sparsity of the DCT of the background, are preserved throughout the learned patches. As the deeper layers carry higher semantically information regarding the images we aim to recover the learned foreground of the model with respect to the deeper convolutional layers.

Based on our hypothesis we present the Signature Activation method. A gradient free non-linear method to generate activation maps that encapsulate the nature of the forward pass, i.e. heatmaps that are faithful to the decision making process of the model, and that highlight the learned regions of fixation for the model. Although previous explanation methods have worked with varying success, they do not explicitly consider the inherent properties of the input images. By disregarding the rest of the classes, with class discriminative explanations certain parts of the foreground could be potentially be disregarded as background incurring in information loss. Due to the required transparency and precision needed to deploy Machine Learning algorithms in clinical settings it is desirable to exploit the structure of inputs, while approximating the inner workings of the models. With the clear separation of foreground and background in the angiograms and the sparsity of the foreground being preserved through the CNN with the added emphasis on the lesion due to the learned classification task, we seek to extract faithfully the learned foreground through the last convolutional layer.

### 6.1. Definition of our method

Let $I \in \mathbb{R}^{n \times n}$ be an image and $A^k \in \mathbb{R}^{l \times l \times c}$ the $k$-th convolutional layer of a CNN model evaluated after the evaluation of the forward pass for the input image $I$. For an accurate model we will suppose that $A^k$ abstracts the foreground and the background of the image in order to take the classification decision. For a given channel $s$ we assume

that it can be decomposed as

$$A_s^k = f_s + b_s$$

where $f_s$ carries the relevant or foreground information and $b_s$ consists of the background or irrelevant information for the classification. As the sparsity is preserved through the forward pass, $f_s$ will remain sparsely supported in the standard spatial basis and $b_s$ sparsely supported in the basis of the DCT. The signature for the activation $A^k$ at the channel $s$ is given by

$$\text{Sig}(A_s^k) = \text{sign}(\text{DCT}(A_s^k)). \tag{2}$$

The image signature is applied to the channels of the last convolutional filter and added over. Following the principles of visual saliency outlined by (Itti et al., 1998), an activation map is produced as follows:

$$\text{M}(I) = B * \left( \frac{1}{S} \sum_{s=1}^{S} [\text{IDCT}(\text{Sig}(A_s^k))] \circ [\text{IDCT}(\text{Sig}(A_s^k))] \right),$$

where $B$ represents a bilateral filter [1] (Tomasi & Manduchi, 1998), $*$ is the convolution operation, and $\circ$ is the Hadamard product. After obtaining $\text{M}(I)$, the map is resized to match the dimensions of the original image. One of the key differences with most of the outlined methods is that the above approach does not rely on the input's class. In the case of multi-class classification the proposed approach allows to detect if multiple objects for which the training data posses label, are present in the given input. This signals that the model might need debugging if the objects of interest are not being highlighted in the activation map.

---

[1]A bilateral filter is used to smooth images, similarly as a Gaussian filter, while preserving edges.

# 7. Results

In this section, we provide empirical evidence that: 1) our method works better than other methods at Weakly Supervised Object Localizatio, 2) our method passes the sanity checks for saliency maps, and 3) our method proves effective and aiding in the detections of lesions in coronary angiograms. The implementation of our method is available at: https://github.com/dtak/signature-activation.

## 7.1. Signature Activation outperforms other methods in Weakly Supervised Object Localization

Discriminative regions might only focus in certain aspects of the located objects while not the objects themselves. As Signature Activation approximates the entire fixation of the model, it tends to create maps that encapsulate the entire objects as can be seen in Figure 3. To assess this claim and evaluate the fidelity of Signature Activation we performed Weakly Supervised Object Localization (WSOL) over the the explanations in the multi-class setting. The task consists of creating bounding boxes based on the heatmaps to analyze how well they matched the ground of detecting the objects. We utilized the ConvNeXtLarge architecture as its performance competes with modern vision transformers and the ILSVRC 2017 (Russakovsky et al., 2015) validation set consisting of $50,000$ images with their corresponding bounding boxes by computing the Intersection over Union (IoU) as a measure of overlap. The IoU is a ratio that measures the overlap between two bounding boxes. It is defined as the area of overlap between the two bounding boxes divided by the area of union of the two bounding boxes. The experiment was designed as follows: utilizing Grad-CAM, Grad-CAM++, Eigen-CAM, and Signature Activation we computed heatmaps with each method over the 50,000 images of the ILSVRC 2017 validation set with the ConvNeXtLarge architecture. For each heatmap we generated bounding boxes that matched the number of ground truth bounding boxes by thresholding from $0.01$ to 1 over the binary mask values until the number of boxes matched the number of ground truth boxes. If the IoU measure was greater than 0.5 it was considered as a positive instance and negative otherwise. As the explanations created through the saliency methods are not originally designed to detect objects, we measured the Error Rate for the number of total predictions. The results are reported in Table 1.

An error can be produces due to multiple reasons: it can be that the saliency maps generates more or less patches than there are true bounding boxes, it can also be the case that that the patches are only located in discriminate region that do not encapsulate the entire objects making the IoU smaller than 0.5, or it can be that the generated bounding boxes are simply capturing background information. As can be seen in Table 1, Signature Activation outperforms

| Method | Error Rate |
|---|---|
| Grad-CAM | 0.9244 |
| Grad-CAM++ | 0.9421 |
| Eigen-CAM | 0.6733 |
| Signature | 0.6407 |

*Table 1.* Table that contains the average Error Rate for the task of generating bounding boxes based on the heatmaps generated by computing Grad-CAM, Grad-CAM++, Eigen-CAM, and Signature Activation. A lower Error Rate indicates better results for the used method.

the rest of the methods at detecting the objects over the images. We hypothesize by observing Table 1 and looking at manual examples, such as in Figure 5, that given the discriminate nature of both Grad-CAM and Grad-CAM++, the highlighted regions only corresponded to relevant features and not the complete objects which makes the generated bounding boxes miss the ground truth ones.

## 7.2. Signature Activation passes the saliency Sanity Checks

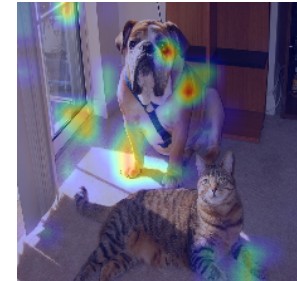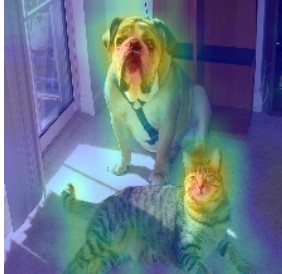

*Figure 5.* Image that contains instances from two classes from Imagenet, Tiger Cat and Boxer Dog, with two different heatmaps. On the left, the heatmap overlay corresponds to the features that are highlighted by Grad-CAM++ with the highest probability corresponding (incorrectly) to *bullmastiff*. On the right, the heatmap overlay corresponds to the features that are highlighted by Signature Activation. Both produced with the second to last layer convolutional layer of ConvNeXtLarge. Signature Activation detects the two subjects of importance.

As discussed by Adebayo et al. (2018), some saliency methods are independent of the architecture and are not fit to generate explanations. Starting from the output layer, the Cascading Randomization test examines the robustness of the saliency method by progressively randomizing layers of the neural network. A method that relies on meaningful features should produce different saliency maps as more layers are randomized. The Independent Randomization test, on the other hand, scrutinizes the method's sensitivity to randomization by just randomizing one of the layer's weight at a time while preserving the rest of the architecture as the original one. If the saliency map changes signifi-

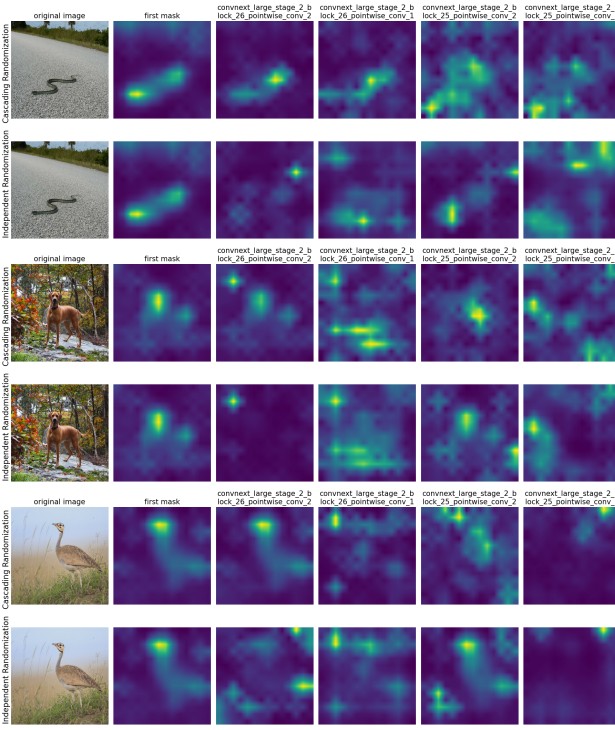

*Figure 6.* Sanity checks for Saliency Activation for two images. The first rows contains the images corresponding to the Cascading Randomization test and the second correspond to the Independent Randomization test. The first image and second image in both rows are the same, the first is the original image and the second is the original Saliency Activation Map. The changes can be seen from left to right, going from the deeper to the shallower layers. By looking at the example we can see how the activation maps change greatly from the original one.

cantly under this condition, it suggests that the method is indeed capturing information from the model's parameters. As can be observed Figure 6, Saliency Activation passes both the Cascading Randomization as well as Independent Randomization tests proposed by Adebayo et al. (2018).

### 7.3. Angiogram Evaluation

A cardiologist manually inspected 790 coronary angiograms with their corresponding masks produced by Signature Activation in order to validate the efficacy of the method. As can be seen in Figures 4 and 7, explanations generated with Signature activation highlight all the relevant regions of the artery. The regions of top most intensity (highlighted in red) tend to align with the location of the lesion. This potentially signals that the method not only detect the vessel completely but it also locates the lesion just like class discriminate methods. The explanation mimics more the process of how a cardiologist make its diagnostics by looking at the artery in its entirety.

## 8. Discussion and Broader Impact

Signature Activation offers a significant shift in approach compared to other prevalent discriminative methods, providing a more holistic view of what the model visualizes during the forward pass. By exploiting the sparsity of the images and the model, this method bypasses the need for intricate architectural manipulation and offers a class-agnostic perspective into the model's decision-making process. The proposed method also leverages the DCT, which enables it to be computationally efficient, thus addressing one of the major challenges faced by many existing methods.

Nevertheless, like all explanation methods, it is not without its limitations. Its local nature means that it may not provide a complete explanation of the entire model. Moreover, its utility might be restricted in tasks where the images lack a clear object of interest.

Despite these caveats, the potential broader impact of this method, especially in clinical settings, could be substantial. In healthcare, the ability to generate more comprehensive and holistic explanations of model predictions could significantly enhance the interpretability of AI-driven diagnostic tools. This could, in turn, lead to increased trust and acceptance from clinicians, who could leverage these insights to make more informed decisions. Moreover, the method's emphasis on exploiting image and model sparsity could potentially align well with other medical imaging tasks apart from angiograms, where relevant features often have a clear foreground-background separation.

## 9. Conclusion

In this work, we provide reasons for why class-discriminative explanations for neural network models provide narrow views of model's decision making process. We propose Signature Activation, a class-agnostic method to generate visual explanation for CNNs that aims to provide a more holistic view of what affects a model's predictions. We provide theoretical justifications for our method as well as empirical evaluations – we verified the fidelity of our explanations through Weakly Supervised Object Localization test, as well as checked that they pass the saliency Sanity Checks. We also explore the potential use of our method in clinical decision making by analyzing a use case in the detection of lesions in Invasive Coronary Angiograms.

## Acknowledgements

This material is based upon work supported by the National Science Foundation under Grant No. IIS-1750358. Any

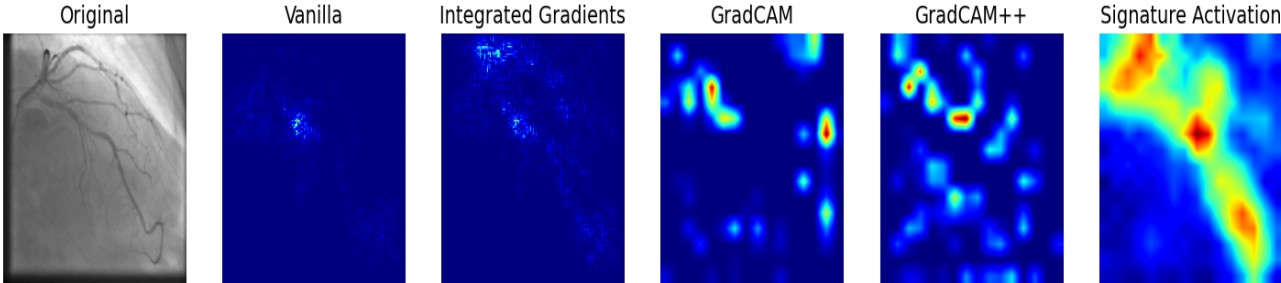

*Figure 7.* One frame from a Left Coronary Angiogram and different masks generated by different methods. Signature Activation effectively highlights the coronary artery, distinguishing it from the background. This ability to emphasize key features, greatly aids in the interpretation process by cardiologists. Specifically, it can help identify potential blockages a critical step in assessing patient heart health using angiogram frames.

opinions, findings, and conclusions or recommendations expressed in this material are those of the author(s) and do not necessarily reflect the views of the National Science Foundation.

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

# A. Appendix 1

Taken from Hou et al. (2012)

**Theorem A.1.** *For a given image $I \in \mathbb{R}^n$ that admits a decomposition as $I = f + b$, the image reconstructed from the image signature approximates the location of a sufficiently sparse foreground on a sufficiently sparse background as follows:*

$$\mathbb{E}\left[\frac{\langle Sig(IDCT(f)), Sig(IDCT(I))\rangle}{\|Sig(IDCT(f)\|_2\|Sig(IDCT(I)\|_2}\right] \geq 0.5,$$

*for $|Support(b)| \leq \frac{n}{6}$.*

