# OpenReview forum: "Signature Activation: A Sparse Signal View for Holistic Saliency"
_ICML.cc/2023/Workshop/IMLH — IMLH 2023 Oral_

### Official Review · Reviewer_zRcm · 2023-06-11
**Comments on IMLH, Submission 93**

**Rating:** 7
**Confidence:** 5

**Review:**

In this study, Signature Activation, a saliency method that generates holistic and class-agnostic explanations for Convolutional Neural Networks' outputs, is introduced by the author. The proposed method exploits image sparsity and provides theoretical justification for its effectiveness. The results of the study demonstrate the potential utility of the proposed method in clinical settings, particularly in aiding the detection of lesions in Coronary Angiograms. The paper is well-written and provides detailed information about the proposed method.

However, I have a concern regarding Figure 2, which illustrates an example where the proposed method highlights both a Tiger Cat and a Remote. Does this imply that the proposed method is suitable for multi-class scenarios? Furthermore, if the objective is solely to detect whether the image contains a cat or not, would the detection of the Remote influence the decision-making process?

---

### Official Review · Reviewer_NT1h · 2023-06-14
**A signal-processing persepctive on saliency detection**

**Rating:** 9
**Confidence:** 3

**Review:**

The authors propose signature activation for saliency detection/visualization for CNN-based image classification. The authors define image/feature signatures as the sign functions of the DCT coefficients, and compute saliency maps based on the inverse DCT of these signatures. The proposed method is tested on coronary angiorams and is shown to yield semantically meaningful saliency maps.

Pros:
1. The idea of leveraging classical signal processing theory is interesting. The overall methodology is neat.
2. The proposed method demonstrates improved qualitative results compared with GradCAM family methods.
3. The task of saliency visualization is clinically relevant.

Cons:
1. The manuscript is a bit wordy and it is therefore difficult to read. It should be made more concise and better-structured.
2. The compared methods are relatively old-fashioned (GradCAM family).
3. It is unclear whether the semantical meaning of saliency maps truly reveals the underlying decision processes of the deep networks or not. More discussions on this potential gap in between are encouraged.

---

### Meta-Review · Area_Chair_eVyS · 2023-06-20

**Recommendation:** Accept (Oral)
**Confidence:** 5

**Metareview:**

The authors designed a novel method called Signature Activation, that can generate holistic and agnostic explanations for CNNs prediction. The experiment designs are clear. The paper garnered high ratings from all reviews, indicating that it meets the high standards required for presentation to the community.

---

### Decision · Program_Chairs · 2023-06-20

Accept (Oral)